# Atypical Hemolytic Uremic Syndrome Associated with BNT162b2 mRNA COVID-19 Vaccine in a Kidney Transplant Recipient: A Case Report and Literature Review

**DOI:** 10.3390/idr17010014

**Published:** 2025-02-11

**Authors:** Eleonora Francesca Pattonieri, Marilena Gregorini, Maria Antonietta Grignano, Tefik Islami, Gioacchino D’Ambrosio, Gianluigi Ardissino, Teresa Rampino

**Affiliations:** 1Unit of Nephrology, Dialysis and Transplants, Fondazione I.R.C.C.S. Policlinico San Matteo, 27100 Pavia, Italy; e.pattonieri@smatteo.pv.it (E.F.P.); t_islami@hotmail.com (T.I.); t.rampino@smatteo.pv.it (T.R.); 2Department of Internal Medicine and Therapeutics, University of Pavia, 27100 Pavia, Italy; 3Anatomic Pathology Unit, Department of Molecular Medicine, University of Pavia and Fondazione IRCCS Policlinico San Matteo, 27100 Pavia, Italy; g.dambrosio@smatteo.pv.it; 4Center for HUS Prevention, Control and Management, Fondazione IRCCS Ca’ Granda Ospedale Maggiore Policlinico, 20162 Milano, Italy; ardissino@centroseu.org

**Keywords:** BNT162b2, COVID-19 vaccination, hemolytic uremic syndrome, acute kidney injury, kidney transplant

## Abstract

**Case Report:** We report a case of a 37-year-old female with kidney transplant, who was admitted at our hospital due to worsening renal function, nephrotic proteinuria, and anemia developed 21 days after the second dose of BNT162b2 COVID-19 vaccine (Pfizer-BioNTech). Laboratory tests revealed hemolytic anemia, thrombocytopenia, and acute kidney injury. Given the clinical picture of Thrombotic Micro-angiopathy (TMA) and severe renal impairment, plasma exchange (PEX) and dialysis were immediately started. Laboratory workup showed low C3 and C4 levels, normal activity of ADAMTS13, and the absence of anti-factor H antibodies. Molecular biology investigations revealed a heterozygous variant in exon 22 (*SCR20*) of the *CFH* gene (*c.3628C*>*T*; *p.Arg1210Cys*) described as an atypical Hemolytic Uremic Syndrome (aHUS) causative mutation. Our patient completed two sessions of PEX followed by eculizumab treatment with hematological improvement but no recovery of renal function. This is the first reported case of aHUS triggered by SARS-CoV-2 vaccination in a kidney transplant patient without recovery of renal function. **Conclusion:** Although rare, clinicians should be aware of possible nephrological complications that may appear after vaccination.

## 1. Introduction

A high mortality and morbidity rate caused by the SARS-CoV-2 pandemic has been recorded since the beginning. It is known that the kidney is one of the main targets of SARS-CoV-2 infection and the availability of vaccines against SARS-CoV-2 has significantly reduced the mortality rate both in the general population and in vulnerable nephropathic patients [1].

Despite the significant benefit associated with containing the pandemic, the potential side effects of these vaccines remain a topic of great concern, particularly in transplant patients [2,3,4,5,6,7,8].

Here we report a case of atypical Hemolytic Uremic Syndrome (aHUS) associated with End-Stage Renal Disease (ESRD) following administration of the second dose of BNT162b2 SARS-CoV2 vaccine (Pfizer-BioNTech) in a renal transplant patient.

Written informed consent was obtained from the patient before describing this clinical case.

## 2. Clinical Case Description

A 37-year-old Caucasian woman received a kidney transplant due to an unknown cause of chronic renal disease. Her medical history records a gastroenteritis, complicated by acute renal damage, anemia, and thrombocytopenia at the age of 10 during a trip to Africa. With the diagnostic tools available at the time, it was not possible to distinguish between hemolytic uremic syndrome (HUS) and atypical HUS. Once the patient’s renal function had improved, no further hematological and biochemical tests were needed.

Later, in 1997, blood tests revealed advanced renal failure, and the patient underwent a kidney transplant. The follow-up for chronic antibody-mediated rejection (cABMR) begins in 2019 at Policlinico San Matteo, Pavia. cABMR was treated with Extracorporeal Photopheresis (EP) which stabilized the renal function [creatinine 2.2 mg/dL, eGFR (CKD-EPI) 26.6 mL/min, urea 124 mg/dL, proteinuria 3.1 g/24 h, and normal C3 and C4 values].

The immunosuppressive treatment consisted of cyclosporine, mycophenolic acid, and steroids and also included tenofovir, to control HBV viremia, and erythropoietin to treat hyporegenerative anemia. In April 2021, she was admitted to our hospital due to asthenia and hypertension that did not respond to standard treatment.

On admission, blood tests showed an abrupt decline of renal function (creatinine 3 mg/dL, GFR was 8.6 mL/min), nephrotic proteinuria (5.97 g/day) and glomerular red blood cells in urine sediment, anemia (hemoglobin 8.6 g/dL), high D-dimer levels (3062 mg/mL vs. normal range 599 mg/mL), low haptoglobin levels (6 mg/dL, normal range 34–200 mg/dL), high LDH levels (408 mU/mL vs. normal range 125–220 g/mL), total bilirubin (2.14 mg/dL), indirect bilirubin (1.61 mg/dL), and reduced serum C3 levels (68 mg/dL vs. normal range 75 mg/dL) and C4 levels (6.5 mg/dL vs. normal range 10–40 mg/dL). Leukocyte and platelet counts were within the normal range, and autoimmunity tests were negative. HCV-Ab was detected in the patient’s serum, but no viremia. HbsAg and HIV-Ab, as well as the PCR COVID-19 test on the nasopharyngeal swab, were negative. An ultrasound examination of the urinary tract and kidneys revealed no abnormalities.

It is important to emphasize that the patient had received the first and second doses of the COVID-19 Pfizer-BioNTech mRNA vaccine 68 days and 21 days prior to hospitalization, respectively. During hospitalization, renal function deteriorated rapidly within a few days (creatinine 4.73 mg/dL, eGFR CKD-EPI 10.99 mL/min, urea 291 mg/dL), anemia worsened (Hb 6.7 g/dL), and platelet count decreased dramatically (35,000/uL); anti-PF4 antibody and COOMBS tests were negative. The peripheral blood smear showed schistocytes; microbiological tests, including SHIGA toxin, were negative.

The clinical picture of acute kidney injury (AKI) and the biochemical data were suggestive of hemolytic microangiopathy. Therefore, we performed a percutaneous kidney biopsy and a complete laboratory investigation for Thrombotic Micro-Angiopathy (TMA).

The TMA investigation included analysis of serum complement levels (including CH50 and AP50), performed using the Wieslab Complement System kit (Euro-Diagnostica, Malmö, Sweden), ADAMTS13 activity, anti-ADAMTS13 antibody, factor H, anti-factor H antibody, and genetic testing for mutations in complement regulatory or effector genes.

The biopsy revealed diffuse thickening of the glomerular capillary walls, basement membrane duplication, mesangial matrix expansion, the collapse of some glomerular capillaries and microvascular thrombosis. These findings were consistent with aHUS superimposed on chronic rejection [9].

Two plasma exchange filtration (PEX) sessions were performed on alternate days without clinical benefit. Hemodialysis was started due to anuria and clinical signs of volume overload.

After vaccination against meningococci (Menveo and Bexsero, GSK, Baranzate, Italy), pneumococci (Prevenar 13, Pfizer, Milan, Italy and Pneumovax 23, MSD, Rome, Italy), and *Haemophilus influenzae* (Hiberix, GSK), the anti-C5 monoclonal antibody eculizumab (900 mg) was administered. At the same time, prophylaxis for invasive meningococcal infections was started with ceftriaxone (2 g daily for 15 days).

After the third administration of eculizumab, we observed an increase in hemoglobin and platelet count, but no improvement in bilirubin, haptoglobin, and renal function. Molecular biology examinations revealed normal levels of CH50 (>70%) and AP50 (>30%), normal activity of ADAMTS13, the absence of anti-factor H antibodies, but a heterozygous variant in exon 22 (*SCR20*) of the *CFH* gene (*c.3628C>T*; *p.Arg1210Cys*) which was described as the causative mutation for atypical Hemolytic Uremic Syndrome (aHUS). The diagnosis of aHUS was made.

The patient was discharged after improvement in anemia (hemoglobin 10 g/dL), platelet (87,000/uL), and persistently high LDH levels (637 mU/mL), total bilirubin (2.5 mg/dL), and indirect bilirubin (0.8 mg/dL).

Complete normalization of hemoglobin, platelet count, and hemolysis indices occurred approximately 2 months after discharge. At the same time, we observed a partial recovery of diuresis, but not of renal function. Consequently, we discontinued ECP treatment, which has been shown to be effective in improving and slowing the progression of renal failure associated with cAMBR at our center [10].

At present, after three years, the patient performs three dialyses per week. No further episodes of aHUS have occurred and she has discontinued eculizumab treatment. She has been added to the waiting list for a second renal transplant.

The diagnostic algorithm applied to this case is summarized in Figure 1.

## 3. Discussion

Our case report describes a rare aHUS that developed in a patient with a pathogenic complement variant after the second dose of Pfizer-BioNTech’s mRNA-based COVID-19 vaccine (BNT162b2). A wide range of kidney disease has been reported following the administration of vaccines, particularly mRNA vaccines. These include both recurrent and de novo glomerulonephritis, in particular IgA nephropathy and minimal change disease [11,12].

In our case, there is a close timely correlation between vaccination and aHUS. However, it cannot be denied that complement dysregulation caused by *CFH* gene mutations, combined with damage to the vascular endothelium by cABMR, contributed significantly to aHUS.

In fact, both cABMR and aHUS share similar pathogenic mechanisms, including endothelial dysfunction and dysregulation of the complement activation pathway (cAP). Moreover, the pre-existing low GFR due to chronic kidney disease may have contributed to irreversible kidney damage [13,14].

The molecular pathways of aHUS activated in our patient are illustrated in Figure 2.

aHUS is a rare but life-threatening complication. Fortunately, treatment with complement blockers has changed the outcome and prognosis of patients with aHUS. Early administration of eculizumab, a C5 monoclonal antibody, leads to improvements in hematologic and systemic manifestations, including renal function, even in patients requiring dialysis [15].

In contrast, renal function may be incompletely restored, especially in those with a history of kidney disease and previous aHUS/TMA [16].

To our knowledge, this is the first case of aHUS caused by anti-SARS-CoV-2 vaccination leading to irreversible kidney failure in a transplanted patient [17,18,19,20,21,22,23,24,25,26,27,28,29,30,31,32,33]. Twenty-one cases of aHUS following anti-SARS-CoV-2 vaccination have been reported, two of which were in kidney transplant recipients [28,30] (see Table 1).

In one case similar to ours (patient with chronic rejection and a mutation in the CFH gene), aHUS treatment led to partial recovery of renal function [30].

However, the timing of aHUS onset after vaccination was variable. It occurred within the first week after the first dose of vaccine (median onset after 5 days and min–max: 2 h–40 days). In 47.6% of cases, the disease manifested after the first dose, in 38% after the second dose, and in 14.4% after the third dose. As shown in Table 1, kidney biopsy was performed in only 42% of patients, probably due to the high risk associated with the procedure given the low platelet count. Histological findings were suggestive of aHUS and showed microvascular thrombosis as well as interstitial and glomerular inflammation. In most cases, the disease prognosis was good and signs and symptoms resolved in most patients treated with anti-complement drugs. Relapse of aHUS occurred in 21% of cases, and only one patient required long-term treatment with anti-complement drugs [34].

The use of eculizumab before and after transplantation is effective in preventing aHUS. The use of eculizumab in secondary hemolytic uremic syndrome is controversial. Recent data support the restrictive use of eculizumab in carefully selected aHUS cases, although close monitoring for relapse after discontinuation of the drug is emphasized [35,36,37,38,39,40].

In our case, we discontinued eculizumab and did not observe any relapse. Actually, our patient is waiting for a second renal transplant. The long-acting C5 monoclonal antibody ravulizumab is now approved for the treatment of aHUS, and allows a reduction in the dosing frequency and an improvement in quality of life in patients with aHUS; therefore, the therapeutic approach may change. However, new strategies for novel drugs for complement blockade in atypical hemolytic uremic syndrome are currently under investigation [41].

## 4. Conclusions

In conclusion, although, aHUS following anti-COVID-19 vaccination is rare, it is a serious event that can lead to loss of renal function and, in some cases, to the start of dialysis treatment, even in healthy subjects. The therapeutic approach for patients at risk for loss of renal function, such as those with chronic kidney disease and/or mutations in complement genes, should be based on a risk–benefit ratio. Anti-complement therapy can be administered together with vaccination. In our case, we did not know that the patient already had aHUS, but it is now plausible that the gastroenteritis episode with acute renal failure at the age of 10 years was indeed aHUS. In patients with known aHUS and/or chronic kidney disease, it is advisable to carefully monitor renal function, urinary sediment, and complete blood count in order to recognize and treat this serious complication in the early stage.

## Figures and Tables

**Figure 1 idr-17-00014-f001:**
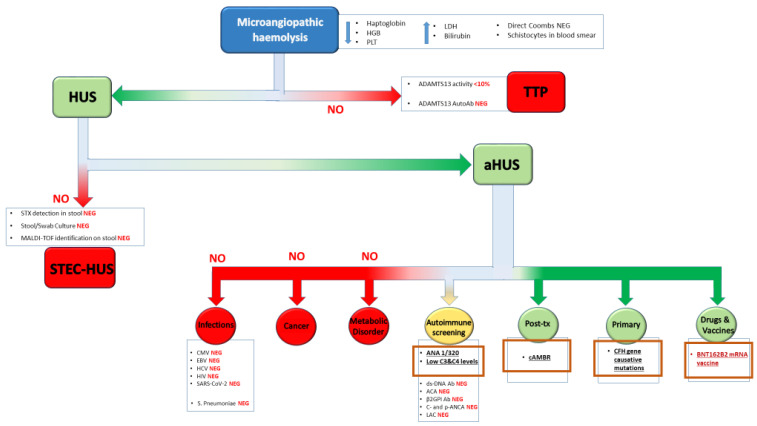
Diagnostic workup. The red “NO” indicates the excluded diagnostic hypotheses. “HGB”: hemoglobin; “PLT”: Platelets; “LDH”: Lactate DeHydrogenase; “Ab”: Antibody; “NEG”: Negative; “TTP”: Thrombotic Thrombocytopenic purpura; “HUS”: Hemolytic Uremic Syndrome; “STX”: Shiga-like ToXin; “STEC-HUS”: Shiga Toxin-producing Escherichia Coli-associated Hemolytic Uremic Syndrome; “aHUS”: atypical Hemolytic Uremic Syndrome; “CMV”: Cytomegalovirus; “HCV”: Hepatitis C Virus; “EBV”: Epstein-Bar Virus; “HIV”: Human Immunodeficiency Virus; “SARS-CoV-2”: Severe Acute Respiratory Syndrome CoronaVirus-2; “S. Pneumoniae”: *Streptococcus Pneumoniae*; “ANA”: AntiNuclear Antibody; “ds-DNA Ab”: double stranded DNA Antibody; “ACA”: Anti-Cardiolipin Antibody; “β2GPI Ab”: β2 Glycoprotein I Antibody; “c- and p-ANCA”: cytoplasmic and perinuclear AntiNeutrophil Cytoplasmic Antibody; “LAC”: Lupus AntiCoagulant; “cAMBR”: chronic active Antibody-Mediated Rejection.

**Figure 2 idr-17-00014-f002:**
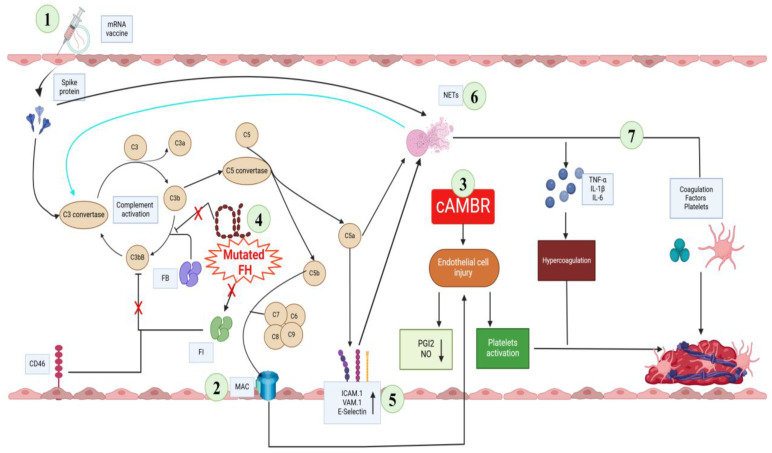
Molecular pathways of aHUS in our case. (1) The mRNA vaccine activates the complement system, leading to the formation of the (2) Membrane Attack Complex (MAC) which causes endothelial cell injury, a decrease in antithrombotic substances (PGI2 and NO), and platelet activation. (3) Chronic vascular rejection (cABMR) makes the endothelium more susceptible to thrombus formation. (4) The mutation of *CFH*, a key protein in the regulation of the complement system, causes an overproduction of the anaphylatoxin C5a. C5a stimulates both (5) the expression of adhesion molecules on endothelial cells and (6) the activation of neutrophils with the release of NETs. (7) NETs enhance the state of hypercoagulation indirectly through the release of inflammatory cytokines and direct binding to coagulation factors and platelets. Created with BioRender.com.

**Table 1 idr-17-00014-t001:** Summary of demographic and clinical characteristics of aHUS post COVID-19 vaccination.

Authors	Age (years)/Gender	Medical History	Vaccine (Which Dose)	Onset Time	Clinical Manifestations	Urine Test	Blood Test	Contributive Cause of TMA	Renal Biopsy	Treatment	Outcome	Others
Ferrer et al. [17]	54/F	Pulmonary tuberculosis, spontaneous abortion	ChAdOx1 (first dose)	5 days	Malaise, abdominal pain, myalgia, vomiting, oliguria, hypertension, AKI.	Hemoglobinuria, 1 mg/dL; UPRO, 100 mg/dL, haematuria.	HGB, 10.3 g/dL; PLT, 50 × 109/L; Scr, 5 mg/dL.	Low C3.	Not performed.	PEX × 10, steroids, hemodialysis, eculizumab.	R (1 month) Scr, 2 mg/dL.	-
Schmidt et al. [18]	60/F	None	BNT162b2 (first dose)	14 days	Discomfort.	ND	HGB, 7.1 g/dL; PLT, 42 g/L; Scr, 2.1 mg/dL.	Pathogenic variant of CFH gene, reduced plasma levels of factor H.	Yes. Glomerular thrombotic microangiopathy.	PEX.	R Renal function, improved.	-
Spasiano et al. [19]	17/F	None	BNT162b2 (first dose)	14 days	Fever, dyspnea, AKI, hypertension, myocarditis.	ND	HGB, 8 g/dL; D-dimer, 2.7 ng/mL; Scr, 7.9 mg/dL.	Pathogenic variant of MMACHC gene.	Yes. Thrombi in capillaries lumen.	Transfusions, hemodialysis, steroids, eculizumab.	NR (2 months). Oliguric, renal replacement treatment, improved after hydroxocobalamin treatment.	-
Lim et al. [20]	69/F	Diabetes mellitus	AZD1222 (first dose)	5 days	Hyperazotemia, thrombocytopenia, weakness, gastrointestinal discomfort.	PCR, 2.7 mg/dL	HGB, 8.5 g/dL; PLT, 38 × 109/L; Scr, 3.7 mg/dL.	ADAMTS13 activity, 68.9%; Shiga toxin result, negative.	Yes. Glomerular intracapillary fibrin deposition with entrapped cellular debris.	None.	R (8 weeks) Scr, 0.65 mg/dL; PCR, 1.0 mg/dL	-
Moreno-Torres et al. [21]	27/F	IgA deficiency, pauci-symptomatic COVID-19	BNT162b2 (first dose)	1 day	Fever, digital ischemia, abdominal pain, hypertension, AKI, DIC.	ND	ND	aCL, aB2GP-1, anti-phosphatidyl-serine/prothrombin antibodies.	Not performed due to transfusion-related acute lung injury.	Hemodialysis, steroids, hydroxychloroquine, low molecular weight heparin.	NR Peritoneal dialysis and will receive renal transplantation.	APLS.
Luiz et al. [22]	39/F	HUS	ChAdOx1 (first dose)	3 days	Nausea, vomiting, epigastric pain, haematuria.	Haematuria.	HGB, 8.8 g/dL; PLT, 80 × 109/L; Scr, 2.2 mg/dL; BUN, 92 mg/dL.	Pathogenic variant of CD46 gene.	Not performed.	PEX × 9, platelet transfusion, steroids	R	Relapsed.
Bouwmeester et al. [23]	26/F	Family history	ChAdOx1 (first dose)	2 days	Fever, dark urine, ongoing epistaxis, nausea.	ND	ND	Pathogenic variant of C3 gene and MCPggaac.	Not performed.	Eculizumab.	R	-
	58/F	Family history	ChAdOx1 (second dose)	3 days	Headache, vomiting, dyspnea on exertion, hypertension, petechiae,	ND	ND	Pathogenic variant of C3 gene.	Not performed.	Hemodialysis, eculizumab.	R	-
	12/M	None	BNT162b2 (second dose)	1 day	nausea, abdominal pain, jaundice, dark, urine, oliguria, petechiae, hypertension.	ND	ND	Pathogenic variant of C3 gene.	Not performed.	Eculizumab.	R	-
	53/M	Antiphospholipid syndrome	BNT162b2 (second dose)	40 days	Fatigue.	ND	ND	Pathogenic variant of C3 gene and MCPggaac.	Not performed.	Eculizumab.	R	-
	57/F	Crohn’s disease	BNT162b2 (second dose)	10 days	Hypertension, diarrhea.	ND	ND	Pathogenic variant of C3 gene and MCPggaac.	Not performed.	Eculizumab.	R	Relapsed.
Tawhari et al. [24]	38/M	None	AZD1222 (first dose)	7 days	Dyspnea, weakness, fatigability, edema.	UPRO, 3.9 g/L; haematuria, 18/HPF.	HGB, 6.7 g/dL; Scr, 10.9 mg/dL; BUN, 43.8 mg/dL.	ADAMTS13 activity, 68.9%; Shiga toxin result, negative.	Yes. Glomerular fibrin thrombi.	Hemodialysis, PEX, FFP, steroids, RTX.	R Scr, 5.1 mg/dL.	ESRD in chronic HD.
Rysava et al. [25]	21/F	Idiopathic epilepsy, ovary resection	mRNA vaccine (second dose)	1 day	Sclera hematomas, thrombocytopenia, AKI, edema, hypertension.	UPRO, 10 g/d.	HGB, 7.8 g/dL; Scr, 2.4 mg/dL.	Pathogenic variant of CFH gene and CD46 gene, low C3.	Yes. Glomerular thrombi.	Steroids, FFP, PEX × 15, eculizumab, ravulizumab.	R (4 months) TMA symptoms, disappeared; renal function, improved.	-
Aku et al. [26]	43/M	None	ChAdOx1 (second dose)	2 h	Chills, fever, generalized body aches.	UPRO, 2 +, haematuria.	HGB, 10.7 g/dL; PLT, 134 × 109/L; Scr, 1.41 mg/dL; BUN, 20.4 mg/dL.	ND	Not performed.	Steroids.	R (16 days).	-
Coric et al. [27]	49/M	None	BNT162b2 (second dose)	10 days	Rhabdomyolysis, haematuria, scanty urination, thrombocytopenia.	Haematuria.	PLT, decreased; Scr, 12.1 mg/dL; BUN, 106.1 mg/dL.	ND	Not performed.	Heparin, Clexane, PEX, hemodialysis.	R Scr, 1.4 mg/dL; BUN, 28 mg/dL.	First dose vaccine, AstraZeneca.
Sogbein et al. [28]	48/M	Renal transplant, TMA after renal transplant, diabetes mellitus	BNT162b2 (second dose)	7 days	AKI.	ND	Scr, 4.27 mg/dL.	ND	Yes. Diffuse lymphocytic interstitial inflammation, peritubular capillaritis, and C4D-negative.	PEX, IVIG, steroids, MMF, sirolimus.	R Scr, 2.9 mg/dL.	Relapsed.
Claes et al. [29]	38/F	Two doses of BNT162b2 5 months ago, contraceptive agent	mRNA-1273 (third dose)	1 day	Headache, malaise, nausea, diarrhea, hypertension, AKI.	ND	HGB, 9.1 g/dL; PLT, 57 × 109/L; Scr, 6.4 mg/dL.	Pathogenic variant of C3 gene and CD46 gene.	Yes. Glomeruli and arterioles thrombi.	PEX × 7, hemodialysis, eculizumab.	R (3 months). Scr, 1.04 mg/dL.	-
Aigner et al. [30]	67/M	Renal transplant, aHUS after renal transplant, cAMBR	mRNA-1273 (fist dose)	10 days	Dyspnea.	ND	Scr, increased.	Pathogenic variant of CFHR1, CFHR3 genes, CFH-H3 gene, and CD46 gene.	Yes Glomerular thrombotic microangiopathy.	Ravulizumab.	R Scr, 2.5 mg/dL.	-
Roldão et al. [31]	54/F	None	ChAdOx1 (first dose)	5 days	Malaise, hypertension, Vomiting, Low urine output, thrombocytopenia, anemia, and AKI.	Hemoglobinuria and proteinuria.	LDH, increased; negative Coombs’s test; undetectable haptoglobin; schistocytes.	Normal ADAMTS13 activity; Shiga toxin result negative; deletion in homozygosity of CFHR3/CFHR1.	Not performed.	PEX × 10, hemodialysis, eculizumab.	R	-
Chen et al. [32]	ND	HUS	ND (third dose)	ND	ND	ND	ND	ND	ND	ND	ND	Double dose of booster.
Moronti et al. [33]	67/M	Hypertension, p-ANCA vasculitis	mRNA-1273 (third dose)	10 days	Anuria, fatigue, anorexia, and nausea.	ND	Scr, 21.23 mg/dL.; BUN, 409 mg/dL; HGB, 10.1 g/dL; PLT, 132 × 10^9^/uL; Haptoglobin <8 mg/dL; D-dimer > 4 ug/mL; p-ANCA, 394.9 UA/mL; schistocytes.	None.	Glomerular thrombotic microangiopathy.	PEX, hemodialysis, steroids, eculizumab.	R Oliguric, hemodialysis, HUS symptoms disappeared.	ESRD in chronic HD; eculizumab treatment maintained.

“AKI” = acute kidney injury; “APLS” = antiphospholipid syndrome; “BUN” = blood urea nitrogen; “DIC” = disseminated intravascular coagulation; “ESRD” = End Stage Renal Disease; “F” = female; “FFP” = fresh frozen plasma; “HD” = hemodialysis; “HGB” = hemoglobin; “HPF” = High Power Field; “HUS”= hemolytic uremic syndrome; “IVIG” = intravenous immunoglobulin; “LDH” = Lactate Dehydrogenase; “M” = male; “MMF” = mycophenolate mofetil; “ND” = data not available; “NR” = patient did not responde to HUS treatment; “p-ANCA” = Perinuclear anti-neutrophil cytoplasmic antibodies; “PEX” = plasma exchange; “PLT” = platelets count; “R” = patient responded to HUS treatment.

## Data Availability

Data are contained within the article.

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
