# Peer review of "Atypical Hemolytic Uremic Syndrome Associated with BNT162b2 mRNA COVID-19 Vaccine in a Kidney Transplant Recipient: A Case Report and Literature Review"

_2036-7449, 2025, doi:10.3390/idr17010014_

Round 1
Reviewer 1 Report
Comments and Suggestions for Authors
The authors should evaluate the possibility of a second kidney transplant; this intervention might be done protecting the patient with available complement inhibitors since she carries a Factor H variant considered causative of disease.
This possibility might be included to improve the discussion of the article.
Author Response
Dear Editor,
Thank you for giving us the opportunity to submit a revised draft of our manuscript titled
“ATYPICAL HEMOLYTIC UREMIC SYNDROME ASSOCIATED WITH BNT162B2 MRNA
COVID-19 VACCINE IN A KIDNEY TRANSPLANT RECIPIENT: A CASE REPORT AND
LITERATURE REVIEW” to Infectious Disease Reports. We appreciate the time and effort that you and the reviewers have dedicated to providing your valuable feedback on the manuscript. We are grateful to the reviewers for their insightful comments on our paper. We have been able to incorporate changes to reflect most of the suggestions provided by the reviewers. We have highlighted the changes within the manuscript.
Here is a point-by-point response to the reviewers’ comments and concerns.
Comments from Reviewer 1
Comment 1: The authors should evaluate the possibility of a second kidney transplant; this intervention might be done protecting the patient with available complement inhibitors since she carries a Factor H variant considered causative of disease. This possibility might be included to improve the discussion of the article.
Response: The reviewer suggestion is right and we are totally in agreement with it. The patient is actually in waiting list for the second transplant. We thank you for pointing this out and as suggested by the reviewer, we have revised the manuscript adding this information to line 120 and to lines 211-212 of the discussion.

Reviewer 2 Report
Comments and Suggestions for Authors
This is an interesting, detailed case report “Atypical Hemolytic Uremic Syndrome Associated With Bnt162b2 Mrna Covid-19 Vaccine In A Kidney Transplant Recipient: A Case Report And Literature Review” by Eleonora Francesca Pattonieri et.al. Authors reported one of the few cases of aHUS which was triggered after administration of COVID-19 vaccine, and this study is important to the scientific community. Overall, the manuscript was well written, and the conclusions made are well-founded with presented results. However, there are FEW PLACES in the manuscript in which the presentation must be improved as noted below
Lane 22: “Laboratory workup showed low C3-C4 levels” Is the C3-C4 represent the ratio of both?
Lane 70: What is the serum C4 levels in the patient?
Lane 79: Insert comma after “dramatically (35000/ul)”.
Lane 85-86: What are the CH50 and AP50 values of the patient? Was CH50 and AP50 assays are performed with kit?
Lane 91: Reference for the statement “These findings were consistent with aHUS superimposed on chronic rejection”.
Lane 116: Suggestion: It would have been better to represent patients’ serum parameters in a table format.
Lane 156: Insert comma after “a key protein in the regulation of the complement system” or rewrite the sentence.
Lane 174: Does patients represented in reference 27 and 29 has kidney failure?
Lane 176: Table 1: Several abbreviations are used in table 1. What is “R” in outcome lane.
Author Response
Dear Editor,
Thank you for giving us the opportunity to submit a revised draft of our manuscript titled
“ATYPICAL HEMOLYTIC UREMIC SYNDROME ASSOCIATED WITH BNT162B2 MRNA
COVID-19 VACCINE IN A KIDNEY TRANSPLANT RECIPIENT: A CASE REPORT AND
LITERATURE REVIEW” to Infectious Disease Reports. We appreciate the time and effort that you and the reviewers have dedicated to providing your valuable feedback on the manuscript. We are grateful to the reviewers for their insightful comments on our paper. We have been able to incorporate changes to reflect most of the suggestions provided by the reviewers. We have highlighted the changes within the manuscript.
Here is a point-by-point response to the reviewers’ comments and concerns.
Comments for Reviewer 2
Comment 1: Line 22. “Laboratory workup showed low C3-C4 levels” Is the C3-C4 represent the ratio of both?
Response: Thank you for pointing this out. A change was made to line 23 of the abstract in order to make the sentence more clear. Specifically the laboratory workup did not reflect the C3/C4 ratio, but rather their individual values.
Comment 2: Line 70. What is the serum C4 levels in the patient?
Response: Thank you for pointing this out. In lines 71 and 72, the level information for C4 has been added.
Comment 3: Line 79. Insert comma after “dramatically (35000/ul)
Response: Thank you for pointing this out. The sentence was corrected, inserting comma in the line 81.
Comment 4: Line 85-86. What are the CH50 and AP50 values of the patient? Was CH50 and AP50 assays are performed with kit?
Response: Based on the reviewer’s suggestion the manuscript has been revised to include information about the kit in lines 88-89 and the values of CH50 and AP50 in line 105.
Comment 5: Reference for the statement “These findings were consistent with aHUS superimposed on chronic rejection”
Response: Thank you for the suggestion, we have added the reference to line 94 of the manuscript
Comment 6: Suggestion: It would have been better to represent patients’ serum parameters in a table format.
Response: Thank you for pointing this out. As suggested we created the table, but, due to the numerous serum parameters of the patient at different times, the table appeared unclear, therefore we decided not to present it.
Comment 7: Lane 156. Insert comma after “a key protein in the regulation of the complement system” or rewrite the sentence.
Response: Agree. As suggested by the reviewer, we have inserted comma in the line 156.
Comment 8: Does patients represented in reference 27 and 29 has kidney failure?
Response: Yes, both studies included patients with kidney failure.
Comment 9: Table1. Several abbreviations are used in table 1. What is “R” in outcome line.
Response: We agree with the reviewer that there isn’t clarity in the table. Thank you for pointing this out. To facilitate the interpretation of the table, we have included a list of abbreviations below.
